# Between support and scepticism: Health professionals' perceptions of a nutrition education program promoting low-carbohydrate, high-fat diets in under-resourced South African communities

Georgina Pujol-Busquets [1,2]*, Kate Larmuth[1,3,4], Christopher C. Webster[1,3,4], James Smith[1,3,4], Ahtisham Younas [5]*, Sergi Fàbregues [2,6], Anna Bach-Faig[2]

1 Division of Physiological Sciences, Department of Human Biology, University of Cape Town, Cape Town, South Africa, 2 NUTRALiSS Research Group, Faculty of Health Sciences, Universitat Oberta de Catalunya (UOC), Barcelona, Spain, 3 International Federation of Sports Medicine (FIMS) Collaborative Centre of Sports Medicine, HPALS, University of Cape Town, Cape Town, South Africa, 4 Research Center for Health through Physical Activity, Lifestyle and Sport, Division of Physiology, Department of Human Biology, Faculty of Health Sciences, University of Cape Town, Cape Town, South Africa, 5 Faculty of Nursing, Memorial University of Newfoundland, St. John's, Newfoundland and Labrador, Canada, 6 Department of Psychology and Education, Universitat Oberta de Catalunya (UOC), Barcelona, Spain

* ay6133@mun.ca (AY); georgipbg@uoc.edu (GPB)

## Abstract

The burden of non-communicable diseases (NCDs) continues to rise, emphasizing the need for effective dietary interventions. Programs such as *Eat Better South Africa (EBSA)* advocate for low-carbohydrate high-fat (LCHF) dietary choices, especially in disadvantaged communities. However, the adoption of such approaches among healthcare professionals remains contentious. This qualitative study explores healthcare professionals' perspectives on nutrition and the EBSA program, drawing on 16 in-depth individual interviews with physicians, nurses, and dietitians from False Bay Hospital, Groote Schuur Hospital, and a primary health clinic in Hout Bay, Western Cape, South Africa. Thematic analysis of the interviews revealed four main findings. Healthcare professionals lacked confidence in their nutritional knowledge, and while many were familiar with the LCHF diet, opinions varied regarding its sustainability and health implications. Concerns were raised about the high fat content and affordability of LCHF foods. Professionals acknowledged the value of group support in behaviour change, as promoted by EBSA, but expressed reservations about its strong emphasis on LCHF diets. Key challenges identified for patients included poverty, cultural beliefs, limited education, and access to nutritious foods. The findings highlight a reliance on traditional dietary advice, with uncertainties about the feasibility and affordability of LCHF diets. These findings offer novel insights into the complexities of implementing community-based dietary interventions in South Africa, with implications for policy and practice.

**Data availability statement:** The full transcripts and raw data from this study cannot be shared publicly due to ethical and legal restrictions. The research involved interviews with healthcare professionals discussing potentially sensitive information related to their workplace, patients, and professional practices. The approved protocol, which was reviewed by the Human Research Ethics Committee of the University of Cape Town (HREC REF 512/2019), did not include provisions for public data sharing. Participants consented to participate in the study under the condition that their full transcripts and identifying information would remain confidential and would not be made publicly available. However, researchers who meet the criteria for access to confidential data may request access to de-identified datasets by contacting the Human Research Ethics Committee of the University of Cape Town (Hub-PhysiologicalSciences@uct.ac.za) or the P.I. Georgina Pujol-Busquets (georgipbg@uoc.edu).

**Funding:** This study received funding from The Noakes Foundation and The National Research Foundation of South Africa. The funders had no role in study design, data collection and analysis, decision to publish, or preparation of the manuscript.

**Competing interests:** The authors have declared that no competing interests exist.

## Introduction

South Africa's public health system faces major challenges, including long wait times, resource shortages, poor management, inadequate infection control, and workforce deficits —exacerbated by inequitable resource distribution, rapid urbanization, and the dual burden of infectious and non-communicable diseases (NCDs) [1–9]. Within this strained system, the healthcare workforce is crucial to advancing universal healthcare goals, particularly in addressing South Africa's complex disease challenges, both infectious and non-communicable [2,9–12]. Despite its critical role in health, nutrition remains poorly integrated into healthcare training, leading to inconsistent advice and under prepared professionals. Yet globally, nutrition is increasingly recognised as central to disease prevention and management, with dietary interventions often matching or exceeding pharmacotherapy in effectiveness and cost-efficiency [13–16]. This is a global concern, as nutrition profoundly influences health outcomes [17–20]. International efforts are bridging gaps in nutrition education, supported by evidence of the effectiveness of dietary and physical activity interventions, which often rival pharmacological treatments in efficacy, safety, and cost-effectiveness [13,16,21–24]. Studies consistently show that nutrition interventions reduce the incidence and severity of diet-related non-communicable diseases (NCDs), improve patient outcomes, reduce healthcare costs, and enhance the well-being of healthcare professionals, especially in acute care settings [25–30].

Various nutritional education initiatives are active in low socio-economic areas of South Africa to address the burden of NCDs, with *Eat Better South Africa (EBSA)* being one prominent program [31]. EBSA promotes a low-carbohydrate high-fat (LCHF) dietary approach, with an emphasis on nutrient-dense, minimally processed foods [31]. The LCHF approach contrasts with the South African Food-Based Dietary Guidelines, which emphasise balanced diets with staple whole grains, moderate fat intake, and a heavy focus on plant-based foods [32]. EBSA aims to empower individuals, particularly women, through dietary education, believing in the transformative power of education on health behaviours and overall well-being [31]. LCHF diets are supported by some clinicians and professional societies as effective treatments for diabetes and obesity management [33]. However, this approach is contentious, with conflicting findings polarising expert opinion and potentially causing confusion among healthcare professionals and patients [34]. Despite these controversies, carbohydrate restriction for managing type 2 diabetes is supported by international consensus guidelines [35,36]. International consensus guidelines cautiously support carbohydrate restriction for type 2 diabetes [35,36], yet historical precedents —such as William Banting's 19th-century dietary polemics— highlight the enduring controversy surrounding this approach [37].

In focus group discussions involving previous participants of EBSA programs, participants noted differences between the dietary guidance provided by healthcare professionals, which aligns with a Western Standard diet in their communities, and the EBSA recommendations [38–40]. Some healthcare professionals who were initially sceptical of the EBSA LCHF diet shifted towards supporting it, as they witnessed positive health outcomes in their patients. A mixed methods evaluation of the EBSA program showed significant improvements in metabolic health markers among participants after six months, including weight loss, reduced waist circumference, and

improved glycaemic control, although adherence remained a challenge due to socioeconomic factors [40]. Additionally, an earlier study reported that program participants felt empowered through nutritional education, but required ongoing support to sustain dietary changes [38]. A qualitative community assessment exploring women's willingness to participate in an EBSA program highlighted both interest in and concerns about the affordability of LCHF diets, as well as challenges related to dietary transition [39]. These findings indicate that while EBSA has had measurable success in improving metabolic health, practical barriers to long-term adherence remain, necessitating better integration with healthcare professionals and community resources.

While some health professionals may support LCHF diets for certain individuals or specific health conditions, others may have reservations based on concerns about long-term health effects, individual variability, sustainability, adherence, and potential nutrient deficiencies [41,42]. Although the EBSA program has shown positive outcomes in improving dietary behaviours and metabolic health among participants, healthcare professionals hold varying opinions on its dietary approach, highlighting the need to explore their perspectives in greater depth. This article aims to explore healthcare professionals' perspectives on the EBSA program, particularly focusing on the effectiveness and sustainability of the LCHF diet in managing NCDs within South Africa. While the primary objective is to examine how healthcare professionals perceive the effectiveness of the EBSA program and its dietary recommendations, this study also investigates the barriers and facilitators that influence their willingness to support or integrate EBSA dietary guidance into their practice. Furthermore, it assesses how professional training, healthcare system constraints, and exposure to evolving nutrition science shape the attitudes of these healthcare professionals toward LCHF diets and nutrition education. It examines the alignment and potential conflicts between the nutritional advice given by professionals and the guidelines promoted by EBSA, identifying gaps in nutrition education and training. This article also explores the broader public health implications of integrating such dietary recommendations and seeks to provide insights for refining the EBSA program to better meet the health needs of South Africa's diverse population.

## Materials and methods

This is a qualitative study using in-depth individual interviews (IDIs) with health professionals from health care facilities that would service current and future EBSA programs. The research team included experts in qualitative research and nutrition science.

### Setting

Participants were selected from two tertiary institutions, namely False Bay Hospital (Fish Hoek, Cape Town) and Groote Schuur Hospital (Observatory, Cape Town), and a primary health clinic in Hout Bay (Hout Bay, Cape Town). False Bay Hospital is a public hospital that was opened in 1965 in Fish Hoek, a town approximately 30 kilometres south of Cape Town. The hospital has approximately 80 beds, and its facilities include a 24-Hour Emergency Unit. It is the only district hospital in the South Peninsula Health District of the Metro Region and serves many communities, including Simonstown, Masiphumelele, Ocean View, Clovelly, and many other surrounding areas. Groote Schuur Hospital is one of Cape Town's largest government-funded academic hospitals and was officially opened in 1938. The hospital has almost 900 beds and it is renowned as the hospital where the world's first human-to-human heart transplant was performed in 1967 by Christiaan Barnard and his team [43]. The Hout Bay health clinic non-profit organization works mostly in an informal settlement called Imizamo Yethu and provides nutrition and health education to the community. Hout Bay is a seaside town in the Western Cape Province situated in the Cape Peninsula, 20 kilometres south of Cape Town.

### Participants

To be eligible for the study, participants had to be health professionals; work in and see patients from the communities where EBSA has conducted a nutrition education program or a community where EBSA is planning to run future

programs; and be able to understand and speak English. Community health clinics were contacted by email or phone to identify potential participants. A researcher explained the study to interested participants, either via phone or at a scheduled meeting, and if they met the inclusion criteria, they were provided with a study information sheet and the informed consent form. An interview visit was then scheduled at a convenient time and place for the participants. The participants met individually with a researcher for approximately thirty minutes to an hour in a private venue within the participating health facilities, ensuring both privacy and accessibility, where they signed the informed consent form and were interviewed in English. Although participants were selected from some facilities serving communities where EBSA had run programs, this did not guarantee their direct involvement in or awareness of the program itself. Additionally, including professionals from communities where EBSA was planning future programs allowed the study to capture prospective opinions on the program, offering insight into potential facilitators and barriers before implementation.

## Procedures

The IDIs were conducted between 29th of November and 21st February 2020 and lasted, on average, 30 minutes. The number of participants is consistent with recommendations in the qualitative research literature, which suggests that meaningful insights can be gained from smaller, information-rich samples [44]. According to this literature, the first five to six IDIs generally produce the majority of new data, and the majority of concepts are identified within the first 10 interviews. As such, new participants were interviewed until saturation was reached [44]. However, as outlined by Braun and Clarke, we acknowledge that the concept of data saturation is debated, with an emphasis on depth of interpretation rather than a fixed threshold for thematic exhaustion [45]. Participants were interviewed on a one-on-one basis to explore their understanding and perceptions of chronic disease, nutrition, LCHF diets, and the EBSA program. A semi structured interview guide (see SI1 – Supporting Information File 1) was used which included both open-ended and probing questions as a guide for the discussion, although questions were not necessarily asked verbatim. Probing questions were asked only if the subjects had not already been covered.

The interview guide was developed based on a review of the literature on healthcare professionals' perceptions of dietary interventions, particularly low-carbohydrate diets. It was piloted with two health professionals (not included in the final sample) to ensure the relevance, clarity, and cultural appropriateness of the questions. The final guide followed a semi-structured format, allowing for follow-up questions and probing to explore emerging topics in depth. It covered the following key areas: healthcare providers' perceptions of their role in community health; major health concerns identified in their communities; views on NCDs such as obesity, diabetes, and cardiovascular disease; the type of dietary advice offered at their healthcare facilities; perspectives on the role of nutrition in chronic disease prevention and treatment; their understanding of local dietary patterns; and their awareness and impressions of the EBSA programme and the LCHF diet, including views on the feasibility of implementing such interventions in their communities.

All interviews were audio-recorded, with the permission of the participants. The interviews were conducted by GPB and saved in a password-protected folder. The recordings were transcribed by a professional transcription service (Cyber Transcription, South Africa) and the accuracy and editing of the transcriptions were double-checked by an external qualitative researcher for rigour and appropriateness. Each transcribed interview was compared with the corresponding audio to ensure that the transcription had been done in context and captured the interview as accurately as possible.

## Data analysis

Braun and Clarke's approach to thematic analysis [46] was followed using NVivo 12 software (QSR International, Melbourne, Australia). The 2006 version of Braun and Clarke's thematic analysis was used. This version is characterised by its epistemological flexibility and openness to the use of codebooks, a feature that has since been abandoned by the authors in more recent iterations of this analytical method [45].

The analysis began with CCW and GPB reading the transcripts several times, summarising them, and making notes on ideas and potential codes. These two researchers then each inductively developed a codebook based on the summaries and notes generated by reading the transcripts in the previous step. Then, they discussed their individual codebooks, merged them into one codebook, and independently coded three randomly selected interviews using separate NVivo projects. CCW and GPB then met again to compare their respective coding, and discuss and resolve disagreements by consensus. The content and structure of the final codebook was finalised through discussions involving ABF, SF, and KL, to ensure consistency and clarity in the coding framework. Once agreement on the codebook was reached among the members of the research team, the remaining interviews were divided into two groups of eight, with each group assigned to GPB or CCW. GPB and CCW held regular meetings to discuss their interpretations, resolve discrepancies, and critically engage with the data. These collaborative discussions allowed for the refinement of the coding framework and the development of sub-themes and themes, ensuring consistency in analytical decisions. The themes developed were then visually organised by GPB using thematic maps, which supported further analysis and interpretation. During and after the interviews, the research team systematically recorded observations and took field notes, including contextual insights and paraverbal cues such as tone, pauses, and emphasis. Although this data was not formally coded, it was used in the process of interpreting and understanding the participants' narratives and in the development of the themes.

The following step involved naming the themes and defining the main concepts of each theme, resulting in a total of four themes. These themes were shared with ABF, SF, KL and JS to confirm their relevance to the research questions. The fact that, as explained above, two researchers (CCW and GPB) were involved in the analysis of the data, and that the different steps of the analysis were reviewed by the other members of the research team, helped to ensure that all the themes were identified and that the data was interpreted in a consistent way, avoiding the possibility of a premature closure of the analysis, limiting the potential effect of bias in the adoption of a singular point of view, and promoting the overall quality of the analysis. We did not use member checking for practical and methodological considerations. Logistically, time and resource constraints prevented us from following up with participants. Methodologically, we adopted an approach that recognises the dynamic and contextual nature of participants' perceptions, which may shift over time and are not necessarily fixed or unique. Participant feedback is shaped by the circumstances in which it is elicited and does not constitute a definitive test of validity [47].

Finally, the last step, carried out mainly by GPB, consisted of producing a written report of the study findings. Quotes from the qualitative data were selected to illustrate the themes identified in the data, ensuring that each captured the essence of the theme. The study adhered to the Consolidated Criteria for Reporting Qualitative Research (COREQ) guidelines to ensure that the methods and findings were reported in a transparent and rigorous manner [48].

### Ethical considerations

The Human Research Ethics Committee (HREC REF 512/2019) of the University of Cape Town approved the study protocol and informed consent procedures. All procedures were performed in accordance with the 1964 Helsinki Declaration and its later amendments. Informed consent was obtained from all individual participants involved in the study. The health facilities of the study consented to the interviews being conducted at their practices.

### Results

The participants recruited for the study were from a variety of general practices and socio-economic localities. A total of 16 healthcare professionals participated in the study, including 7 medical physicians, 7 nurses, and 2 dietitians. To ensure transparency and traceability in the reporting of quotes, each participant was assigned a code reflecting their profession and participant number (e.g., MPh.1 for Male Physician 1, FN.1 for Female Nurse 1, FD.1 for Female Dietitian 1). The average age of the participants was 37.1 ± 9.8 years. A majority of participants (n = 13) worked only in the public sector,

two participants practiced in both the public and private sectors, and one participant practiced only in the private sector. Four themes (Table 1) were developed from the analysis.

## Theme 1: Foundations of dietary understanding

Health professionals recognised that they are not that well-informed about nutrition and its role in disease prevention. They felt constrained by time, stating that while nutritional advice is important for disease prevention, they struggle to incorporate it into clinical consultation. However, they acknowledged that nutrition plays a crucial role in promoting healthy-diet behaviours and preventing diet-related chronic diseases. There was a general sense of confusion regarding nutrition, as participants described encountering contradictory dietary information from different sources, making it difficult to provide clear guidance to patients. As a result, they tend to use many "catchphrases" and their own statements of beliefs. This is effectively based on what has traditionally been promoted as healthy behaviour: small portions, reducing red meat, eating fruit and vegetables, drinking water, reducing dietary fats, avoiding unsaturated fats (for instance, removing fat and skin from chickens) and practising physical activity.

**Table 1. Themes developed from IDIs with health professionals.**

| Theme | Sub-themes | Definition | Quotes |
|---|---|---|---|
| Theme 1. Foundations of dietary understanding | 1.1.Knowledge of nutritional guidelines 1.2.Confidence in nutritional counselling | Refers to their understanding of dietary principles, nutritional requirements, and the role of food in promoting health and preventing disease among the population. This knowledge encompasses various aspects including nutritional guidelines, assessment, dietary counselling and nutrition education. | "Studies have shown that with diabetes even if you increase the medication or you optimize the medication, at the end of the day it's the diet that kind of makes or breaks the disease" (FPh.2). "We basically follow the protocols of the National Health Department (…) but it's outdated and it's not up to speed with what's really happening" (MPh.4). "I definitely don't say you need to cut it out completely [sugar] (…) I understand we all do enjoy sweet things here and there" (FD.1). |
| Theme 2. LCHF: Controversies and beliefs | 2.1. Awareness of LCHF diets 2.2. Concerns over high-fat content | Understanding of health professionals around the recommendations of LCHF diets which depend on their training, experience and personal beliefs. | "I was on [a LCHF diet], but because I didn't have a lot of support (…) I just left it then, but it's actually a good diet (…) I didn't feel so tired" (FN.5). "If you eat high fat things your cholesterol is going to go up" (FPh.3). "I don't advocate a LCHF diet but I'm supportive of what my patients want to try and then I monitor them closely. I watch their cholesterols, their triglycerides, their blood pressures" (MPh.2). "I wouldn't encourage it [LCHF diet], but if a patient really wanted to do it and ask for my advice, I would try to guide them on it yes, for sure" (FD.2). |
| Theme 3. EBSA's role in community nutrition | 3.1. Perceptions of group support 3.2. Uncertainty about program sustainability | Health professionals' understanding and perceptions of the EBSA program promoting LCHF within under-resourced communities which encompass EBSA's group support approach, diet recommendations and program's sustainability. | "When you're in a group, then there's more support from one another" (FN.3). "I think it's a great initiative [EBSA], very necessary because if it's successful, it will definitely reduce the disease burden in our community and that impacts on everything" (FPh.3). "If the people don't have the funding, they're not going to be able to buy it [EBSA's recommended food]" (FPh.7). |
| Theme 4: Understanding socio-economic determinants of health | 4.1. Affordability and food access 4.2. Cultural and gender norms shaping diet | Health professionals' understanding of the barriers that under-resourced patients face in accessing healthcare and maintaining good health. Their views on these barriers are informed by their direct interactions with patients, as well as their broader understanding of social determinants of health. | "Beggars can't be choosers". While they [people from low-income communities] might know that what they're eating is not good for them, they don't have the means to put that into place" (MPh.1). "It's a very big cultural thing to be big [body size]. There's a stigma around it if you're skinny and you're losing weight, you are becoming sick" (FPh.6). "I think in certain cultures, women don't have a lot of say in what gets done, so if the man comes home and he says "he wants this sort of meal" that's what she must prepare". (FPh.3). |

LCHF: Low carbohydrate high fat; EBSA: Eat Better South Africa; M: Male; F: Female; Ph: Physician; N: Nurse; D: Dietitian.

Nurses and physicians often lacked confidence in their own nutritional knowledge and reported feeling uncertain when discussing dietary concepts. Rather than advising patients themselves, many preferred to refer them to dietitian, especially if they felt that the patient required specific nutritional advice. They cited contradictory information from science, media, and government policies as key sources of confusion. Health professionals highlighted this inconsistency as a challenge for both themselves and their patients, contributing to public uncertainty about nutrition. A few health professionals attributed their lack of confidence in providing nutrition advice to insufficient training in both their formal education and continued professional development. Few participants referenced scientific evidence when justifying dietary recommendations. Therefore, while science was often cited as a source of conflicting information, it was rarely directly referenced by participants. Their confusion appeared to be from inconsistent public health messaging, institutional guidelines, and media influence rather than direct interaction with scientific literature.

Despite not keeping up with new developments in nutrition science, participants agreed that more clinical trials are needed to reassess current dietary guidelines and challenge dietary stigmas, particularly regarding dietary fat and cholesterol levels. Dietary guidance often relied on simplified "catchphrases" (e.g., "balanced diet", "portion control", "sustainable diet") rather than evidence-based specifics. While dietitians emphasized moderation, physicians and nurses were more likely to label foods as "good" or "bad", focusing on reducing sugar, salt, and processed foods. Notably, healthcare professionals rarely referenced scientific literature directly, attributing confusion to conflicting public health messaging rather than engagement with primary research. These "catchphrases" can be defined as statements that are repeated without any justification or attempt to explain them, as if they were obvious. Another notable pattern in participant responses was the use of absolutes when discussing dietary beliefs, often presenting certain nutritional claims as self-evident truths. Phrases such as "obviously", "we all know", and "we have known for years" indicated that health professionals regarded certain dietary messages as unquestionable.

There was also some consistent advice that was tailored to specific conditions. For instance, low salt for hypertension and portion control and exercise for obesity. The professionals agreed indirectly that healthy food improves immunity; therefore, human immunodeficiency virus (HIV) patients should be encouraged to eat healthy foods because their immune system is compromised. In general, the cohort thought that good advice would be to promote physical activity, cut out simple sugar, stop or reduce smoking, lower salt intake, reduce alcohol, not eat spicy food, remove the skin off chicken, remove fat from red meat, and zero sugar sodas instead of regular ones. Although there were differences in how dietitians, physicians, and nurses approached nutrition advice, there was a shared tendency to frame dietary guidance in practical rather than prescriptive terms. While dietitians emphasized moderation and portion control, physicians and nurses were more likely to label specific foods as problematic. Foods considered detrimental to health included those high in sugar and fat, convenience foods, large portions, starchy foods, sugary drinks, processed snacks, fast foods, and overly salty foods. Examples of such foods were described as: instant or convenience food, inexpensive and easy to access, served in large portions, high-starch content, sugary drinks, "chippies and sweeties", junk food, high-calorie food, salty food, "braai-ed" [cooked on a barbecue] meat and more. Healthy alternatives were described as vegetables, fruits, small portions, oats, three balanced meals a day, healthy snacks (such as apples or whole-wheat crackers), and avoiding processed foods and sauces. Some professionals also recommended low-fat options, choosing chicken over red meat and reducing carbohydrate intake. Some mentioned that they advised their patients to eat food in their natural state, not too overprocessed, in moderation and avoid sauces. In general, the advice on practicing physical activity seems to be non-specific and is used when explaining healthy living behaviours and promoting health.

## Theme 2: LCHF controversies and beliefs

Almost all the health professionals had heard of the LCHF diet, though their understanding varied significantly. For those unfamiliar with it, "A low-carbohydrate high-fat diet is a way of eating that restricts the amount of carbohydrates and increases the number of healthy fats that you eat to optimize your health and nutrition". Some professionals recognised

LCHF diets as the Banting or Tim Noakes diet, as they said that these terms have been popularised over the past few years in South Africa, particularly through social media. While some participants correctly described LCHF as a diet promoting whole, unprocessed foods, others had limited knowledge about the specific foods recommended in the diet. When discussing their opinions on LCHF diets, health professionals frequently used vague, generalized statements such as "long-term health" (Female physician) or "not sustainable" (Female dietitian), indicating uncertainty or a lack of critical engagement with the diet. A significant number of participants had personal experience or first-hand knowledge of LCHF diets, with many describing positive outcomes such as needing to eat less, having increased energy levels, and having reduced cravings. However, some stopped following LCHF due to challenges such as a lack of family support or the perception that the diet was expensive. Those who had initially been sceptical of LCHF often noted that it contradicted their formal university nutrition education, particularly regarding the role of fruit, fibre, and starch in the diet.

Public controversy surrounding LCHF further reinforced scepticism among some health professionals, leading them to express concerns and share their reservations with patients considering or already following the diet. Their primary concern focused on the high fat content of LCHF diets and whether the diet was financially sustainable for patients from low-income communities. Many professionals believed that LCHF foods are expensive, making it difficult for patients to maintain a long-term diet. Some were willing to support patients who chose to follow LCHF, but their support was often conditional on closely monitoring patients for potential negative effects. While international consensus guidelines recognize carbohydrate restriction as an effective approach for managing type 2 diabetes, scepticism remains among some health professionals, particularly regarding its long-term sustainability and applicability in under-resourced settings. Some health professionals questioned whether LCHF could effectively treat metabolic disorders, largely due to ongoing controversy surrounding the diet and the lack of full scientific agreement on its long-term effects. Their main concerns included the sustainability of the diet, its affordability, the high intake of meat, the restriction of carbohydrates, and, most notably, its fat content owing to the perceived link between fat intake and cholesterol levels. Some professionals acknowledged that their views on cholesterol and dietary fat might be outdated, attributing this to limited access to updated nutrition science.

Health professionals also expressed concerns about patients' ability to fully understand and follow LCHF correctly, fearing that misinformation and a lack of nutritional literacy could lead to poor adherence or unintended health risks. Some worried that patients would struggle to transition to a low-carb diet, particularly without professional guidance and ongoing monitoring. When asked whether they would introduce LCHF to treat or prevent metabolic conditions, most stated that they would support patients who chose it but would not actively advocate for the diet due to the concerns outlined above. This cautious stance led many professionals to adopt a "wait-and-see" approach, where they would monitor patients rather than actively encourage the diet.

Some alternative reasons for their reluctance included the belief that LCHF is too restrictive and that vegetables should not be excluded from a healthy diet. Many professionals also believed that patients lacked adequate knowledge about their own health before adopting LCHF, further reducing their willingness to endorse the diet. The two dietitians interviewed highlighted adherence as a key challenge, stating that while LCHF may be effective for some individuals, an individualized dietary approach may be more appropriate. As a result, they did not advocate for LCHF as a universal dietary solution, emphasizing instead that dietary interventions should be tailored to each patient's needs.

### Theme 3: EBSA's role in community nutrition

Some of the interviewees were familiar with the EBSA program, but the majority were introduced to it for the first time during the interview. Those who were aware of EBSA had patients who had participated in the program. For clarity, the EBSA program was defined to all participants as: "Eat Better South Africa is a nutrition education program for people from under-resourced communities that promotes low-carbohydrate high-fat diet or Banting diet to prevent or treat metabolic conditions. It consists of groups of people that meet every week for an educational session for a period of six weeks".

Many professionals viewed group support as essential for dietary interventions, stating that changing lifestyle behaviours is easier when done collectively rather than individually. Some participants saw EBSA as a positive initiative for reshaping their dietary mindset and recognised its potential to alleviate pressure on healthcare facilities by reducing the incidence of diet-related chronic diseases. However, concerns about the EBSA program often overlapped with broader concerns about LCHF diets, particularly in terms of sustainability, affordability, and accessibility in low-income communities. Health professionals also emphasized the need for structured nutrition education programs tailored for healthcare workers themselves. Many expressed that they lacked the necessary tools and training to provide effective dietary guidance and stated that a program similar to EBSA, designed for health professionals, would help them better address nutrition-related diseases in their practice. While the majority of participants stated that they would support patients participating in EBSA, they specified the limits of their support, indicating that they would monitor but not necessarily promote the program's dietary approach.

Despite these reservations, group support was considered the most valuable aspect of the EBSA program. Many professionals agreed that individual behaviour change is difficult and that patients are more likely to succeed when they receive support from their peers. Some participants valued the educational aspect of EBSA, stating that if more at-risk individuals joined similar community-based nutrition programs, the burden of chronic diseases on hospitals could be reduced. Concerns about EBSA primarily mirrored existing concerns about LCHF diets, with professionals questioning whether the diet could be adapted to the financial and food accessibility challenges of under-resourced communities. Some interviewees stressed the importance of tailoring dietary recommendations to local food prices and availability, ensuring that any dietary intervention remains affordable and practical for long-term adherence. Several health professionals also provided suggestions on how EBSA could better address food affordability and sustainability, indicating a willingness to engage with the program if these issues were explicitly considered in its implementation.

### Theme 4: Understanding socioeconomic determinants of health

Health professionals broadly acknowledged the challenges that people living in under-resourced communities face, demonstrating awareness of the socioeconomic barriers to healthy living. Many participants referred to specific communities such as Ocean View and Masiphumelele, which are often highlighted in the media for high crime rates, violence, and safety concerns. Some health professionals reside in the same communities as their patients do, providing them with firsthand insight into the rationale behind food choices and the structural challenges that shape dietary behaviours. This theme captures interconnected issues related to food accessibility, affordability, safety, and knowledge. Several health professionals noted that for many individuals, food is not a priority, and that they are in a survival mode. The cost of food was often cited as an issue, emphasizing that healthier foods tend to be more expensive than processed and unhealthy options. Meat was consistently described as expensive, though some also noted that vegetables and other fresh produce were increasingly costly, further complicating efforts to promote healthier dietary choices.

The interviewees cited poverty and lack of resources due to socio-economic circumstances as a challenge to healthy lifestyle choices and changes. According to the information that the participants provided, communities such as Ocean View and Masiphumelele face high levels of unemployment, crime, inadequate housing, and limited access to healthcare and nutrition education. Unemployment appeared to be a major contributing factor to the high level of poverty in these communities. Health professionals also considered limited education a barrier to adopting healthy lifestyle choices, changes, or nutritional interventions. They recognised that nutrition education could have a strong influence on changing unhealthy habits and improving long-term health outcomes. Alcohol abuse was mentioned as a contributing factor to both crime and the reinforcement of unhealthy behaviours, which further complicated adherence to medical and lifestyle recommendations. Compliance with prescribed medication, treatment, and lifestyle advice was identified as a challenge for the patients from the health professionals' perspective.

Health professionals acknowledged that culture and tradition strongly influence dietary habits and perceptions of health, which are often shaped by family and community norms. Cultural stigmatization around body weight and physical appearance was a recurring theme, particularly regarding women. Some participants noted that being slim was sometimes associated with poverty and malnutrition, whereas larger body sizes were perceived as indicators of wealth and well-being. Gender roles within households were also highlighted, as women were often responsible for preparing meals yet had to navigate traditional expectations that limited their decision-making power regarding food choices, health, and financial autonomy. Another key theme was the diverse cultural backgrounds within these communities, which influenced dietary habits and nutritional choices. Some health professionals discussed how migration from rural to urban areas impacted dietary behaviours. In rural settings, individuals often had access to fresh, homegrown food and engaged in more physical activity. However, upon relocating to urban environments, they tended to maintain a high-carbohydrate diet while adopting more sedentary lifestyles, contributing to diet-related health concerns. The transition from rural to urban living was perceived as a challenge, as individuals often lost access to traditional food sources such as home gardening or farming, increasing their reliance on processed and nutrient-poor foods.

Health professionals noted that despite receiving nutritional advice, many patients do not adhere to dietary recommendations, often due to aversion to unfamiliar foods or deeply ingrained eating habits. They identified a preference for ultra-processed foods, explaining that these foods are not only widely available but also enjoyed for their taste, making compliance with healthier dietary guidance more challenging. Some professionals linked this preference to food addiction and eating disorders, highlighting the difficulty in breaking habitual consumption patterns of processed, high-sugar foods. An important cultural aspect raised by professionals was the link between food and social status in under-resourced communities. People with limited financial resources may prioritize purchasing branded, ultra-processed foods as a way to signal wealth or social belonging, even if those foods are nutritionally poor. Professionals observed that certain food choices were not solely based on affordability but also on perceptions of prestige and aspiration, further complicating efforts to promote healthier eating habits.

## Discussion

This study is the first to explore health professionals' perceptions of a community-based program promoting LCHF diets in under-resourced South African communities. The findings reveal a general alignment with conventional dietary guidelines among health professionals, who emphasize the importance of vegetables and fruits while expressing concerns about dietary fats and meat [49]. While physicians and nurses recognized the value of nutritional education in disease prevention, their ability to provide effective dietary counselling was hindered by systemic barriers, including time constraints, competing clinical responsibilities, and inconsistent nutritional messaging. [50]. This reflects a broader issue of confusion and inconsistency in nutritional advice, which can undermine the effectiveness of health promotion efforts.

The study also found that different healthcare professionals approach nutrition advice differently. Physicians admitted to having some nutritional knowledge but lacked confidence in offering dietary guidance, while nurses, who are often the primary healthcare providers, demonstrated more openness to new information and a greater willingness to provide lifestyle counselling, particularly in managing non-communicable diseases (NCDs) [51–55]. However, their ability to provide effective dietary counselling was limited by time constraints, competing responsibilities, and a lack of structured training. An in-depth qualitative investigation with nurses delved into their encounters while counselling diabetes patients on diet, exercise, and smoking cessation [55]. Nurses highlighted challenges like patients' limited knowledge and motivation, as well as their own counselling skills and time constraints, hindering effective lifestyle counselling. Despite these obstacles, the study revealed nurses' readiness to offer weight management advice as part of their professional duties, aligning with previous research findings [54,55]. Dietitians, though fewer in number, adhered more strictly to existing dietary guidelines and were cautious about endorsing LCHF diets due to concerns about their sustainability and potential health risks [56]. While health professionals expressed difficulty in staying updated with advancements in nutrition science, they also relied

heavily on widely accepted dietary principles without necessarily questioning or critically analysing them. Furthermore, many professionals agreed that more clinical trials were needed to reassess current dietary guidelines, reflecting an awareness of the evolving nature of nutrition science. This duality, acknowledging the need for stronger evidence while continuing to use conventional dietary rhetoric, highlights the complexity of how professionals navigate nutritional knowledge in practice.

Despite the cautious approach of most health professionals, there was an acknowledgment of the potential benefits of LCHF diets in clinical nutrition, especially if patients are closely monitored. However, concerns about practical challenges in implementing diets were prevalent, including the cost and variety of recommended foods, the feasibility of adherence in low-income settings, and the alignment with national dietary guidelines were prevalent [34,57,58]. Health professionals generally viewed these barriers as more pressing than the debate over the diet's health effects, suggesting that practical implementation challenges could be the primary concern. While most health professionals claimed that they would support patients, their approach often inadvertently discouraged them, resembling "backhanded support", explicitly noted in some instances. Their reluctance stemmed from concerns about the diet's alignment with conventional dietary guidelines, a lack of confidence in providing LCHF-specific advice, and uncertainty about its long-term health impacts. Rather than actively endorsing LCHF diets, professionals supported patients out of a desire to respect patient autonomy, a stance known as therapeutic pragmatism, where healthcare providers adopt a supportive role despite personal doubts [59]. This meant that while they outwardly agreed to assist patients following an LCHF approach, their hesitance and underlying scepticism often resulted in a lack of meaningful guidance, leaving patients to navigate dietary changes largely on their own. Alternatively, physicians and nurses sometimes directed patients to dietitians, who, as revealed in interviews, might dissuade them from LCHF diets in favour of a more balanced macronutrient approach.

Some study participants had personal experiences with or first-hand knowledge of LCHF diets from friends or family, which tended to be positive. Interestingly, despite their cautiousness in endorsing or advocating for the diet, some had personally experimented with it and experienced favourable results, reflecting findings from other studies in which individuals reported improved well-being, weight loss, and reduced hunger in the diet [60–63]. When discussing the LCHF diet and introducing the EBSA program as a nutrition and health education initiative promoting this approach, health professionals showed interest in the concept of group support, suggesting potential backing for community-based programs that provide such support. The availability of group support within the program was seen as a catalyst for encouraging dietary behaviour changes. However, this interest did not necessarily translate into full endorsement of the LCHF diet in communities, as concerns persisted about various barriers to implementation, including food costs, accessibility, and conflicting advice from different health workers. While LCHF diets are widely recognized in South Africa, their popularity is largely driven by media visibility and public discourse rather than strong institutional or professional endorsement. Many healthcare professionals remained cautious about formally supporting the diet, reflecting a broader global trend where media-driven interest does not always translate into clinical adoption [60]. This suggests that scepticism about LCHF is not solely a matter of questioning its health implications but rather an uncertainty about how it can be practically adopted in resource-limited settings. Some professionals also noted that hesitation toward LCHF did not come solely from healthcare providers but was also present within the support groups themselves, which were composed of community members, including patients. This reluctance within the groups stemmed from conflicting advice received from different health workers, scepticism about non-traditional diets, and broader uncertainty about the feasibility of LCHF within low-income settings [61–63]. While some professionals acknowledged that the program might be effective in changing perceptions about diet, concerns about the LCHF approach were also raised.

There was a general recognition of the challenging environmental conditions in under-resourced South African communities, with food insecurity emerging as a significant concern due to issues of accessibility and affordability. Therefore, socioeconomic determinants emerged as critical barriers to dietary change. Additionally, although not explicitly highlighted, discussions also touched on women's safety concerns. Cultural influences and traditional dietary habits were frequently

mentioned, echoing previous research [64–66]. These factors were seen as obstacles to both individual and community well-being. Health professionals often framed these challenges as significant barriers to dietary change rather than outright rejecting LCHF as a viable nutritional intervention. This rationalization of limited intervention in nutritional guidance aligns with previous observations [50]. The findings of this study highlight how culture, tradition, and socioeconomic status influence dietary behaviours and perceptions of health, often complicating the effectiveness of nutritional education and dietary interventions. The association between larger body sizes and wealth, as noted by health professionals, aligns with previous research showing that body image norms in some South African communities equate size with prosperity and well-being, while thinness is often linked to illness or poverty [67]. These perceptions, combined with gendered household roles, place women in a position where they are responsible for meal preparation but have limited autonomy in making food choices, reinforcing structural barriers to dietary change [68]. Additionally, the migration from rural to urban areas has disrupted traditional food systems, increasing reliance on processed and commercially available foods, which mirrors broader trends in the African nutrition transition [69].

Health professionals also expressed frustration over patients' low compliance with dietary recommendations, with many preferring familiar ultra-processed foods that are perceived as enjoyable and, in some cases, status symbols. This echoes previous research highlighting the role of food choices as social markers in economically constrained environments, where individuals may prioritize brand-associated foods as symbols of affluence despite financial hardship [70]. These findings echo broader research on nutrition transitions in Africa, where urbanization and economic constraints drive reliance on processed foods [69]. Understanding these complex social and cultural influences is critical for designing effective, context-specific nutrition education programs that resonate with communities while addressing underlying barriers to dietary change. This perspective connects with the notion of "backhanded support" when it comes to assisting patients interested in following an LCHF diet. Health professionals acknowledged that unhealthy foods are often consumed because they are enjoyable [51,56,71–73], linking this to concepts of food addiction and predictors of problematic eating behaviours. However, many professionals believed that economic factors and social influences played a more decisive role than individual dietary preferences in shaping food choices. Notably, healthcare providers, particularly dietitians, appeared hesitant to challenge these preferences among participants.

This study has a temporal limitation, as data collection took place between November 2019 and February 2020, meaning that the findings capture a specific moment in time. Dietary discourse in South Africa may have evolved since then, particularly due to pandemic-related shifts in food access, nutrition policies, and public health priorities. While this study provides valuable insights into health professionals' perspectives, future research is needed to explore how these views may have changed with emerging dietary trends and transition initiatives. Based on the study findings, EBSA could improve their program by offering targeted training for healthcare providers, particularly nurses and physicians, to boost their confidence in recommending LCHF diets. This training should address misconceptions about dietary fats and provide strategies for monitoring patients effectively. Additionally, EBSA could develop resources that consider the socioeconomic challenges and cultural influences in under-resourced communities, offering practical and affordable dietary advice. Enhancing the group support component could also strengthen community involvement and help sustain dietary changes, leading to better health outcomes. This study is a pioneering effort to explore health professionals' views on a community-based program promoting LCHF diets in underprivileged South African communities.

Qualitative interviews revealed that health professionals largely align with conventional dietary guidelines, emphasizing vegetables and fruits while expressing concerns about fats and meat. The diverse perspectives of physicians, nurses, and dietitians offer a comprehensive view, but the uneven sample size, limited facility access, the affiliation of one dietitian with an NGO, and the exclusion of non-English speakers may have limited the diversity of perspectives captured and introduced biases. Future studies should include non-English-speaking health professionals to capture broader cultural perspectives. Additionally, while dietitians are typically the primary providers of dietary counselling, in under-resourced primary care settings, nurses and physicians often take on this role due to staff shortages [52]. This reflects the real-world

task-shifting that is common in South African healthcare, where non-specialist providers must frequently step in to deliver nutritional advice. While dietitians play a central role in dietary counselling, this study included physicians and nurses to capture how nutrition messages are communicated in primary care, aligning with community health models for under-resourced settings. Future research focusing solely on dietitians could provide more specialized insights into LCHF counselling and implementation.

By highlighting the crucial role of nurses in health promotion, this study underscores the need for further research to explore how task-shifting impacts dietary counselling and to address the broader systemic challenges influencing health-care practices. Despite these challenges, the study offers valuable insights into health professionals' attitudes towards nutritional advice and LCHF diets, contributing to discussions on community healthcare and NCD management. Notably, some doctors privately believed in LCHF's potential benefits —often informed more by media than by formal training— but were reluctant to refer patients to dietitians, anticipating that dietitians would discourage LCHF in favour of conventional low-fat advice [13].

## Conclusions

This study provides valuable insights into healthcare professionals' perspectives on a nutrition education program recommending LCHF diets, highlighting key challenges in their understanding and implementation within under-resourced South African communities. The professionals expressed uncertainty around LCHF, which was often shaped by conflicting guidelines, limited exposure to recent evidence, and scepticism about its long-term effects. The findings also reveal significant training gaps in nutrition education, with many professionals, mostly doctors and nurses, lacking confidence in providing dietary guidance beyond conventional recommendations. Additionally, socioeconomic barriers such as food affordability, cultural norms, and structural inequalities further complicate dietary behaviour changes in these communities. Despite these challenges, according to participants, EBSA presents a promising model for integrating nutrition education into healthcare strategies, particularly by offering structured support and practical guidance tailored to local contexts. Future efforts should focus on bridging knowledge gaps, strengthening professional training, and ensuring that dietary interventions are both evidence-based and accessible to those who need them most.

## Supporting information

**S1 File. Semi structured interview guide.**
(DOCX)

## Acknowledgments

We acknowledge the health professionals who volunteered to participate in this study, and Cyber Transcription for transcriptions. Thank you to the hospitals and non-profit organizations for allowing the interviews to be conducted in their facilities.

## Author contributions

**Conceptualization:** Georgina Pujol-Busquets.

**Data curation:** Georgina Pujol-Busquets, Kate Larmuth.

**Formal analysis:** Georgina Pujol-Busquets, Kate Larmuth, Christopher C Webster, James Smith.

**Investigation:** Kate Larmuth.

**Methodology:** Georgina Pujol-Busquets, Christopher C Webster, James Smith, Sergi Fàbregues, Anna Bach-Faig.

**Project administration:** Sergi Fàbregues.

**Resources:** Sergi Fàbregues.

**Supervision:** James Smith.

**Validation:** Ahtisham Younas, Sergi Fàbregues, Anna Bach-Faig.

**Writing – original draft:** Georgina Pujol-Busquets, Kate Larmuth, Christopher C Webster, James Smith, Ahtisham Younas, Sergi Fàbregues, Anna Bach-Faig.

**Writing – review & editing:** Georgina Pujol-Busquets, Kate Larmuth, Christopher C Webster, James Smith, Ahtisham Younas, Sergi Fàbregues, Anna Bach-Faig.

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
