## [Decision Letter · Decision Letter 0]

Apr 13 2025

Dear Dr. Younas,

Thank you for submitting your manuscript to PLOS ONE. After careful consideration, we feel that it has merit but does not fully meet PLOS ONE’s publication criteria as it currently stands. Therefore, we invite you to submit a revised version of the manuscript that addresses the points raised during the review process.

We look forward to receiving your revised manuscript.

Kind regards,

Mickael Essouma, M. D.

Academic Editor

PLOS ONE

Journal Requirements:

4. In the online submission form, you indicated that [Data cannot be shared publicly because of ethical restrictions. Data are available from the first author Georgina Pujol-Busquets on reasonable request.].

Additional Editor Comments:

I have appended some comments in the attached pdf alongside reviewers' comments.

The authors are urged to address all reviewers' comments, except the second comment of Reviewer 2.

Reviewers' comments:

Reviewer's Responses to Questions

**Comments to the Author**

1. Is the manuscript technically sound, and do the data support the conclusions?

Reviewer #1: Yes

Reviewer #2: No

Reviewer #3: Yes

Reviewer #4: Partly

2. Has the statistical analysis been performed appropriately and rigorously?

Reviewer #1: N/A

Reviewer #2: No

Reviewer #3: Yes

Reviewer #4: N/A

3. Have the authors made all data underlying the findings in their manuscript fully available?

Reviewer #1: No

Reviewer #2: No

Reviewer #3: No

Reviewer #4: No

4. Is the manuscript presented in an intelligible fashion and written in standard English?

Reviewer #1: Yes

Reviewer #2: Yes

Reviewer #3: Yes

Reviewer #4: Yes

Reviewer #1: This study explores healthcare professionals' perspectives on nutrition and the EBSA program, drawing on 16 in-depth individual interviews with physicians, nurses, and dietitians from False Bay Hospital, Groote Schuur Hospital, and a primary health clinic in Hout Bay, Western Cape, South Africa. Thematic analysis revealed four main theme- Foundations of dietary understanding, LCHF: Controversies and beliefs; EBSA’s role in community nutrition; and understanding socioeconomic determinants of health. It is nicely written, and I have no major concerns.

I offer the following suggestions for your consideration:

1. While I absolutely agree with the first paragraph, I felt that it didn’t align well with the study performed. Can this be more focussed on the research question? Or, perhaps, just delete it and start with the sentence in line 70 (“Despite nutrition’s…”)

2. What would be useful to a reader not intimately familiar with SA nutrition/health systems would be a brief (just brief, no need for >4 sentences) comparison of EBSA vs. standard dietary guidance in South Africa. Could this be added to provide this context?

3. Was only selecting English speaking participants a major bias?

4. Why is “original language” stated in line 153? Were all interviews not conducted in English, as stated in line 137-38?

5. Was member checking performed? If not, might be good to justify why it was not done.

6. Which profession usually counsels on nutrition in this setting? If it’s dietitians, would having only 2/16 participants as dietitians be a limitation of perspective?

7. Discussion line 448- could you provide an example of what is meant by “backhanded support”?

8. Discussion line 448-450- this is an interesting and possibly loaded interpretation. Could you expand? Is it that the doctors believed in LCHF (despite admitting limited nutrition knowledge), and were reluctant (“sometimes”) to send to dietitians because they feared that dietitians would not provide LCHF advice?

9. Discussion lines 459-464; could you clarify- was the failure to endorse LCHF in communities one of the group, in that the support group itself (was this composed of patients) was not endorsing LCHF? It’s not clear

Reviewer #2: 1. May please add some of the observations in the tabulated form.

2. It would be better if you could convert the qualitative discussion output as quantitative values, which can be statistically analyzed.

3. Sentences may be edited for clarity.

4. The presented conclusion could be part of the discussion, and a new conclusion based on the study conducted may be developed.

5. Was there any control group?

Reviewer #3: Review comments

Title: Between support and skepticism :Health professional's views on low carbohydrate diest in under-resourced South African communities.

The paper is written well except the following gaps;

1.Content or focus: much of professional/expert skepticism is raised against the applicability or practicability of EBSA's LCHF program ,not on its health implications. All the issues under theme 4 are arguments questioning practicability.Thus, it would have been better the research was framed along that direction.

2.Your findings go beyound the scope of your title. Your title is restricted only to Low Carbohydrate Diets, but your findings and discussions are focusing more on the controvercies over High Fat Diets. Look at the folowing on the sources of skepticism;

line 35, 303, 311-312, and 420. These all are stressing on expert caution on High Fat diets than Low Carbohydrates.

3.Unsupported and contradictory statements. For example,

A)health professional opinions on the role of nutritional education, (line 2002-2003 contradicts with (line 421_422).

B)information access of professionals,(line 229 contradicts with line 430).

C)popularity of LCHF diets in south africa (504) contradicts with lack of support and stopping to follow those diets mentioned in the manuscript

D.effectiveness of LCHF, consensus guidelines on LCHF (line 74)Vs lack of scientific consensu on it (309)

E. Venue of meeting interviewee: private venue(line 137) Vs in the health facilities(519)

F.Duration of contact with interviewees: one hour(137) Vs thirty minutes(141)

G.selection of interviwee was done from facilities that run EBSA program nutrition education (113). That means they are expected to be knowledgabe about the program.However, you have written ( line 329-330) that "majority were introduced to (EBSA) program for the first time during the interview"

4) lack of clarity on the role of science: line 309 and 221 which similarly state that "confusing information comes from science". These two contradict with line 226 "few health professjonals made any reference to science". Does science resolve doubts or produces and ignites them? It also begs the question , if most of the experts are not refering science, why and how do they can be confused with scientific evidences on the topic?

5). Tha dat collecton (for this research) conducted in nov 2019 and the corresponding findings might not be of high relevance to adress the new dietary trends, including dietary transition initiatives.

6).inaccurate targeting of the sample population. The right targets should be nutritionists/dieticians. Not only that, they also should not be from private facilities.

Generally, the paper has methodological and content gaps that need to be corrected.

Reviewer #4: I am grateful for the opportunity to review this study, which already has the merit of bringing to light a little-known reality and featuring authors whose backgrounds make the work international.

The study is interesting, but I notice that the interviews were conducted more than 5 years ago, and the bibliography is a lot but very dated. I suggest motivating why it took so long to publish this work and what justifies using interviews from 5 years ago. Health issues and contexts evolve very quickly.

I wonder what is new about a paper that analyses data from 5 years earlier. I would therefore suggest specifying well what this study brings that is new. Furthermore, the bibliography should be updated to the last 10 years, preferably 5 years. Braun and Clarke 2006 can also be updated.

Here are some more suggestions:

Title: Specify in the title the type of research: a qualitative research

Keywords: Redefine keywords by adding for example perceptions, diet... and changing the order of the words: 6. thematic analysis; 3 low-carbohydrate, high-fat diet; 5 South Africa; 4 under-resourced communities; 2. health professionals. 1. perspectives on nutrition

Background:

The EBSA programme is not adequately described and the relevant literature does not support the use of this diet with solid evidence. I suggest that it should be better explained and recent and specific literature on this programme should be included. If it is a new programme, it should be explained in detail.

I would also explain in more detail studies 38-39-40 that are already specifically evaluating this programme.

In this case, perhaps the research problem could be specified before the objective. For example, it could be: The results on the population of this dietetic education are positive (?), but the professionals still have different opinions. What then are the opinions of the professionals on the ESBA education programme?

The objectives of the study are different and do not facilitate consistency between objective, data collection instrument (interview) and analysis. I would propose to select only one objective, the first one, at least as a macro-objective. The others could become sub-objectives.

Material and methods

Since the setting is very detailed, insert a sub-paragraph with ‘setting’. Line 116

Participants. As a rule, participants in the study are also given a consent form for the use of the data. With this consent, anonymised and aggregated data can be disseminated. Has a particular choice been made in this respect?

Inclusion criteria: ‘a community where EBSA is planning to run future programmes’. Why?

Procedures

Line 142: There is the problem of the year of the interviews, but it has already been addressed in the introduction.

Line 144: I have some concerns about the sample and saturation. I prefer to talk about participants, rather than sampling, which is more typical of quantitative research. With respect to data saturation, opinions are still very diverse today. While I do not agree with what has been written, I would not change it as you have cited the literature. (I would add, however, that using thematic analysis by Braun and Clarke, these authors have written several articles on data saturation, always in a reflexive manner).

Line 149. It would be important to explain how the interview was constructed.

Line 154. Ok the deregistration with software, but as you have reviewed the transcripts, I would ask you if you have pointed out the researchers‘ field notes or if you have retained part of the participants’ paraverbal communication, useful elements for the analysis.

Line 159: Table 1: I would suggest moving the detailed, very thorough interview to the annex . In the text insert a summary of the interview, reworked with main subject areas and, for each area, one or two sample questions.

Before analysing the data, you should include which instrument was used for reporting the qualitative research. On the basis of the chosen instrument, a paragraph on the research team could be inserted, specifying the expertise in qualitative research.

Data Analysis

Line 165. You cite Braun and Clarke (2006) and it is correct, but these authors have also written extensively on thematic analysis in very recent years. One could update the literature.

Line 169. The provision of codebooks for qualitative research is very good. I would suggest placing the basic data from which the analysis is developed in a repository linked to the article.

Line 179. Thematic analysis (by Braun &Clarke) is developed into themes and sub-themes. The transition to concepts is more titpical of grounded theory. I would relate the description of the analysis to your chosen model (themes and sub-themes)

The description of the analysis is very detailed and offers many insights into the rigour of the research. However, I would suggest pointing out a reference author with a citation for rigour (by including it as a citation in the analysis) or make a separate paragraph pointing out which dimensions of rigour you specifically considered (with reference author).

Results

Line 199. I would enter the total number of respondents in the introduction of the results. I would then imagine that a code, e.g. numeric, was assigned to each participant, while maintaining the difference in profession ( Cod. Ph. 1 to indicate the first code, e.g. Physician)

I would rework the table of themes, keeping the 4 main themes, but also inserting the sub-themes (which can be derived from the definitions) for each theme. The quotes are too many and too long (they should be an example), so I suggest selecting them and reducing them in length. For example.

Themes Sub-themes quotations

1. Foundations of dietary understanding

1.1. knowledge of nutritional guidelines, assessment.. (e.g.) ‘We basically follow the protocols of the National Health Department and (..) there isn’t really much with regards to prevention, they have information, but it’s outdated (..) (Male physician).

1.2. dietary counselling and nutrition education

Text of Results

The text of the results could also be reworked with themes and sub-themes. It would make comprehension more immediate, in my opinion.

I really appreciate it when the codes are included in the text in full or as parts of the quotes (e.g. lines 247-249, 252..). I would however add the code.

Line 287. ‘For those who did not know about the diet..’ to remedy this, the fact of knowing about this diet could be included in the inclusion criteria.

Line 389. I would not include citations in the results

The results, which are very extensive, go far beyond the professionals' view of the ESBA education programme, which is little or not at all known to the professionals interviewed.

The various problems highlighted in the results concern the ESBA programme only in part, because they then range over the whole aspect of nutrition education, in poor populations for whom the food problem is not the priority, the professionals‘ lack of knowledge, their need for training, and above all the professionals’ doubts as to whether or not this programme is effective and how to do effective nutrition education, and the role of women in African communities.

Even the title, with these results, is no longer explanatory.

I therefore suggest that we seek consistency between the title, objectives and results, selecting results that express clarity on what is achieved with this study.

It seems to me that the main underlining feature at the moment is the confusion of the professionals as to which programme to focus on and the request to be informed about nutrition education so as to be of help to the population.

The conclusions follow from what has been said for the results.

**Do you want your identity to be public for this peer review?** For information about this choice, including consent withdrawal, please see our Privacy Policy

Reviewer #1: No

Reviewer #2: No

Reviewer #3: **Yes: ** Abdelah Alifnur Mohammed

Reviewer #4: No

---

## [Author Response · Author response to Decision Letter 1]

23 Apr 2025

Manuscript Title: Between support and scepticism: Health professionals’ views on Low-Carbohydrate diets in under-resourced South African communities.

New Tile: Between support and scepticism: Health professionals’ perceptions on a nutrition education program promoting Low-Carbohydrate, High-fat diets in under-resourced South African communities.

Manuscript ID: PONE-D-24-55230

Dear Editor and Esteemed Reviewers,

On behalf of all co-authors, I would like to sincerely thank the reviewers for their thoughtful and constructive feedback on our manuscript. We greatly appreciate the time and expertise that went into reviewing our work, and we are confident that the suggested revisions have significantly improved the quality and clarity of our submission. We have carefully considered each comment and have responded in detail below, outlining the changes made to the manuscript and providing clarifications where necessary.

We trust that our revisions have addressed the reviewers' concerns, and we look forward to your further consideration of our manuscript for publication in PLOS ONE.

Sincerely,

Georgina Pujol-Busquets

Reviewers' comments:

Comments to the Author

1. Is the manuscript technically sound, and do the data support the conclusions?

Reviewer #1: Yes

Reviewer #2: No

Reviewer #3: Yes

Reviewer #4: Partly

2. Has the statistical analysis been performed appropriately and rigorously?

Reviewer #1: N/A

Reviewer #2: No

Reviewer #3: Yes

Reviewer #4: N/A

3. Have the authors made all data underlying the findings in their manuscript fully available?

The PLOS Data policy<http://track.editorialmanager.com/CL0/http:%2F%2Fwww.plosone.org%2Fstatic%2Fpolicies.action%23sharing/1/010f019547d21ab6-08e7bc50-c91e-4e43-a149-14affdf0e8f0-000000/uoBneyU2ESJ2opnWfelmiRHTzi2lhcmhawIbFSjISQo=200> requires authors to make all data underlying the findings described in their manuscript fully available without restriction, with rare exception (please refer to the Data Availability Statement in the manuscript PDF file). The data should be provided as part of the manuscript or its supporting information, or deposited to a public repository. For example, in addition to summary statistics, the data points behind means, medians and variance measures should be available. If there are restrictions on publicly sharing data—e.g. participant privacy or use of data from a third party—those must be specified.

Reviewer #1: No

Reviewer #2: No

Reviewer #3: No

Reviewer #4: No

4. Is the manuscript presented in an intelligible fashion and written in standard English?

Reviewer #1: Yes

Reviewer #2: Yes

Reviewer #3: Yes

Reviewer #4: Yes

5. Review Comments to the Author

Responses to Reviewers' Comments

Reviewer #1

This study explores healthcare professionals' perspectives on nutrition and the EBSA program, drawing on 16 in-depth individual interviews with physicians, nurses, and dietitians from False Bay Hospital, Groote Schuur Hospital, and a primary health clinic in Hout Bay, Western Cape, South Africa. Thematic analysis revealed four main theme- Foundations of dietary understanding, LCHF: Controversies and beliefs; EBSA’s role in community nutrition; and understanding socioeconomic determinants of health. It is nicely written, and I have no major concerns. I offer the following suggestions for your consideration:

1. While I absolutely agree with the first paragraph, I felt that it didn’t align well with the study performed. Can this be more focussed on the research question? Or, perhaps, just delete it and start with the sentence in line 70 (“Despite nutrition’s…”)

Response:

We agree that the first paragraph did not fully align well with the study focus. As suggested, the paragraph has been modified, and the Introduction now begins with:

"South Africa's public health system faces major challenges, including long wait times, resource shortages, poor management, inadequate infection control, and workforce deficits—exacerbated by inequitable resource distribution, rapid urbanization, and the dual burden of infectious and non-communicable diseases (NCDs) [1–9].”

2. What would be useful to a reader not intimately familiar with SA nutrition/health systems would be a brief (just brief, no need for >4 sentences) comparison of EBSA vs. standard dietary guidance in South Africa. Could this be added to provide this context?

Response:

We have added a brief comparison between EBSA and standard dietary guidance in South Africa in the Introduction. This contextual framing is now included to orient readers unfamiliar with South African dietary policy.

“EBSA promotes a low-carbohydrate high-fat (LCHF) dietary approach, with an emphasis on nutrient-dense, minimally processed foods [31]. The LCHF approach contrasts with the South African Food-Based Dietary Guidelines, which emphasise balanced diets with staple whole grains, moderate fat intake, and a heavy focus on plant-based foods [32].”

Reference:

32. Vorster HH, Badham JB, Venter CS. An introduction to the revised South African food-based dietary guidelines. S Afr J Clin Nutr. 2013;26(Suppl 1):S5–S12.

3. Was only selecting English speaking participants a major bias?

Response:

We acknowledge this as a limitation and have now explicitly mentioned this in the limitations section at the of the Discussion, noting that the exclusion of non-English speakers may have limited the diversity of perspectives captured.

“The diverse perspectives of physicians, nurses, and dietitians offer a comprehensive view, but the uneven sample size, limited facility access, the affiliation of one dietitian with an NGO, and the exclusion of non-English speakers may have limited the diversity of perspectives captured and introduced biases. Future studies should include non-English-speaking health professionals to capture broader cultural perspectives.”

4. Why is “original language” stated in line 153? Were all interviews not conducted in English, as stated in line 137-38?

Response:

This has been corrected to explicitly state that all interviews were conducted in English. The term “original language” has been removed to avoid confusion.

5. Was member checking performed? If not, might be good to justify why it was not done.

Response:

Member checking was not performed in this study due to both logistical and methodological considerations. Practically, the time and resource constraints inherent in the research timeline made it infeasible to re-engage participants for feedback. However, beyond these constraints, there are also epistemological and methodological concerns associated with the use of member checking in qualitative research of this nature. One common critique, as discussed by Bloor (1997), is that member validation exercises are not neutral or definitive tests of validity, but rather complex social encounters influenced by context, participant subjectivity, and the changing nature of perception. As Bloor demonstrates, members’ reactions to findings are often shaped by the passage of time, evolving perspectives, and the dynamics of the feedback process itself. Participants might reject or revise earlier statements not because the initial data were invalid, but because their current viewpoints differ, highlighting the inherently dynamic and situated nature of perceptions. Furthermore, member checking has been critiqued for its roots in positivist assumptions about data validation—that there is a singular, fixed “truth” that participants can confirm. In interpretive qualitative paradigms, such as the one guiding this study, the researcher acknowledges that data are co-constructed and context-dependent, and that seeking verification from participants can sometimes undermine the complexity and multiplicity of meanings present in qualitative data (Bloor, 1997).

Therefore, the decision not to employ member checking aligns with both practical limitations and a reflexive approach that prioritizes depth of interpretation over post hoc validation by participants.

“We did not use member checking for practical and methodological considerations. Logistically, time and resource constraints prevented following up with participants. Methodologically, we adopted a reflexive approach that recognises the dynamic and contextual nature of participants’ perceptions, which may shift over time and are not necessarily fixed or singular. Participant feedback is shaped by the circumstances in which it is elicited and does not constitute a definitive test of validity [47].”

Reference:

47. Bloor M. Techniques of validation in qualitative research: a critical commentary. In: Miller G, Dingwall R, editors. Context and method in qualitative research. London: SAGE Publications; 1997. p. 37–50.

6. Which profession usually counsels on nutrition in this setting? If it’s dietitians, would having only 2/16 participants as dietitians be a limitation of perspective?

Response:

We have clarified in the limitations section in the Discussion that while dietitians are typically the lead providers of dietary counselling, nurses and physicians in under-resourced primary care settings often deliver nutrition advice due to staff shortages. This sampling choice, therefore, reflects real-world task-shifting common in South African healthcare.

“Additionally, while dietitians are typically the primary providers of dietary counselling, in under-resourced primary care settings, nurses and physicians often take on this role due to staff shortages [52]. This reflects the real-world task-shifting that is common in South African healthcare, where non-specialist providers must frequently step in to deliver nutrition advice. While dietitians play a central role in dietary counselling, this study included physicians and nurses to capture how nutrition messages are communicated in primary care, aligning with community health models for under-resourced settings. Future research focusing solely on dietitians could provide more specialized insights into LCHF counselling and implementation.”

7. Discussion line 448- could you provide an example of what is meant by “backhanded support”?

Response:

This phrase has been clarified in the Discussion to mean that professionals reluctantly supported patients who chose LCHF diets, not because they endorsed the diet themselves, but because they wanted to encourage patient autonomy. This illustrates a phenomenon known as therapeutic pragmatism, where professionals adopt a supportive stance despite personal doubts.

“Their reluctance stemmed from concerns about the diet’s alignment with conventional dietary guidelines, a lack of confidence in providing LCHF-specific advice, and uncertainty about its long-term health impacts. Rather than actively endorsing LCHF diets, professionals supported patients out of a desire to respect patient autonomy, a stance known as therapeutic pragmatism, where healthcare providers adopt a supportive role despite personal doubts [59]. This meant that while they outwardly agreed to assist patients following an LCHF approach, their hesitance and underlying scepticism often resulted in a lack of meaningful guidance, leaving patients to navigate dietary changes largely on their own.”

Reference:

59. Pilnick A, Hindmarsh J, Gill VT, editors. Communication in healthcare settings: policy, participation and new technologies. Chichester: Wiley-Blackwell; 2010. 160 p. ISBN: 978-1-4051-9827-1.

8. Discussion line 448-450- this is an interesting and possibly loaded interpretation. Could you expand? Is it that the doctors believed in LCHF (despite admitting limited nutrition knowledge), and were reluctant (“sometimes”) to send to dietitians because they feared that dietitians would not provide LCHF advice?

Response:

This section has been expanded at the end of the Discussion to explain that some doctors privately believed in LCHF’s potential benefits (often informed by media rather than training), but were reluctant to refer to dietitians because they expected dietitians to discourage LCHF in favor of conventional low-fat advice.

“Despite these challenges, the study offers valuable insights into health professionals' attitudes towards nutritional advice and LCHF diets, contributing to discussions on community healthcare and NCD management. Notably, some doctors privately believed in LCHF’s potential benefits—often informed more by media than formal training—but were reluctant to refer patients to dietitians, anticipating that dietitians would discourage LCHF in favor of conventional low-fat advice [13].”

9. Discussion lines 459-464; could you clarify- was the failure to endorse LCHF in communities one of the group, in that the support group itself (was this composed of patients) was not endorsing LCHF? It’s not clear

Response:

This has been revised to clarify in the Discussion that some professionals felt community-based support groups themselves were hesitant to fully endorse LCHF diets. This stemmed from uncertainty, conflicting advice from different health workers, and broader scepticism about non-traditional diets within these groups.

“Many healthcare professionals remained cautious about formally supporting the diet, reflecting a broader global trend where media-driven interest does not always translate into clinical adoption [60]. This suggests that scepticism about LCHF is not solely a matter of questioning its health implications but rather an uncertainty about how it can be practically adopted in resource-limited settings. Some professionals also noted that hesitation toward LCHF did not come solely from healthcare providers but was also present within the support groups themselves, which were composed of community members, including patients. This reluctance within the groups stemmed from conflicting advice received from different health workers, scepticism about non-traditional diets, and broader uncertainty about the feasibility of LCHF within low-income settings [61-63]. While some professionals acknowledged that the program might be effective in changing perceptions about diet, concerns about the LCHF approach were also raised.”

Reviewer #2

1. May please add some of the observations in the tabulated form.

Response:

Thank you for your suggestion. Table 2 (now Table 1) was developed initially to provide deeper insights into the themes, offering a structured view of health professionals' perspectives with definitions and direct quotes to support the qualitative analysis. By summarizing key findings while maintaining participant perspectives, we think that Table 1 effectively presents observations in a clear, accessible format. We believe it meets the request but welcome further clarification if needed.

2. It would be better if you could convert the qualitative discussion output as quantitative values, which can be statistically analyzed.

Response:

Thank you for the comment. While some qualitative studies may include elements of quantification, we do not consider it appropriate in the context of this particular study. The approach employed is interpretive, where the aim is to explore meaning and complexity rather than to measure frequency.

3. Sentences may be edited for clarity.

Response:

The manuscript has undergone thorough language editing to improve clarity and readability.

4. The presented conclusion could be part of the discussion, and a new conclusion based on the study conducted may be developed.

R

---

## [Decision Letter · Decision Letter 1]

Between support and scepticism: Health professionals’ perceptions of a nutrition education program promoting low-carbohydrate, high-fat diets in under-resourced South African communities

PONE-D-24-55230R1

Dear Dr. Younas,

We’re pleased to inform you that your manuscript has been judged scientifically suitable for publication and will be formally accepted for publication once it meets all outstanding technical requirements.

Kind regards,

Mickael Essouma, M. D.

Academic Editor

PLOS ONE

Additional Editor Comments (optional):

The authors have improved their manuscript. The reviewers' comments and my comments have been addressed, although I did not see an explicit point-by-point response to my comments in the rebuttal letter. Regarding the concerns raised by Reviewer 3, the authors have provided a detailed and explanatory data availability statement (lines 627-638). The authors described the involvement of different authors in the methods section and also provided a credit author statement. Regarding the perspectives of health workers on the EBSA program, there is a theme dedicated to participants (health workers)' perspectives about the EBSA programme in the results section and the authors have also made a comment on participants' perspectives about the EBSA-related LCHF program in the discussion section.

Altogether, the manuscript is scientifically acceptable for publication. However, I was unable to open some links provided in the reference section. I have appended some last edits that can be made when correcting the proof in the document PONE-D-24-55230_R1_Mickael Essouma comments attached to this decision letter.

Mickael Essouma

Reviewers' comments:

Reviewer's Responses to Questions

**Comments to the Author**

Reviewer #1: All comments have been addressed

Reviewer #3: All comments have been addressed

Reviewer #4: All comments have been addressed

2. Is the manuscript technically sound, and do the data support the conclusions?

Reviewer #1: Yes

Reviewer #3: Yes

Reviewer #4: Yes

3. Has the statistical analysis been performed appropriately and rigorously?

Reviewer #1: N/A

Reviewer #3: N/A

Reviewer #4: N/A

4. Have the authors made all data underlying the findings in their manuscript fully available?

Reviewer #1: Yes

Reviewer #3: No

Reviewer #4: Yes

5. Is the manuscript presented in an intelligible fashion and written in standard English?

Reviewer #1: Yes

Reviewer #3: Yes

Reviewer #4: Yes

Reviewer #1: Thank you for carefully considering the review. My comments and suggestions from the review have been addressed adequately.

Reviewer #3: The authors have showed a concerned engagement with the comments and tried to answer questions and address the comments. But I still have the following queries;

1. The issue of data availability; I did not see a plausible justification from the authors

2. The EBSA program vs professional views still needs clarification

3. I also did not feel that all researchers have equally engaged and contributed in the original research work. Here you have to be clear about the division of tasks/roles among researchers

Reviewer #4: I thank the authors for the great work done in reviewing the manuscript. The answers provided to the comments raised by the reviewer are clear, coherent, exhaustive.

**Do you want your identity to be public for this peer review?** For information about this choice, including consent withdrawal, please see our Privacy Policy

Reviewer #1: **Yes: ** Russell Jude de Souza

Reviewer #3: No

Reviewer #4: No

---

## [Editor Report · Acceptance letter]

PONE-D-24-55230R1

PLOS ONE

Dear Dr. Younas,

I'm pleased to inform you that your manuscript has been deemed suitable for publication in PLOS ONE. Congratulations! Your manuscript is now being handed over to our production team.

Kind regards,

on behalf of

Dr. Mickael Essouma

Academic Editor

PLOS ONE